# Prognostic Significance of EGFR, HER2, and c-Met Overexpression in Surgically Treated Patients with Adenocarcinoma of the Ampulla of Vater

**DOI:** 10.3390/cancers16152756

**Published:** 2024-08-03

**Authors:** Se Jun Park, Kabsoo Shin, Tae Ho Hong, Sung Hak Lee, In-Ho Kim, Younghoon Kim, MyungAh Lee

**Affiliations:** 1Division of Medical Oncology, Department of Internal Medicine, College of Medicine, The Catholic University of Korea, Seoul St. Mary’s Hospital, Seoul 06591, Republic of Korea; psj6936@naver.com (S.J.P.); agx002@naver.com (K.S.); ihkmd@catholic.ac.kr (I.-H.K.); 2Cancer Research Institute, College of Medicine, The Catholic University of Korea, Seoul 06591, Republic of Korea; 3Department of General Surgery, The Catholic University of Korea, Seoul St. Mary’s Hospital, Seoul 06591, Republic of Korea; gshth@catholic.ac.kr; 4Department of Pathology, College of Medicine, The Catholic University of Korea, Seoul St. Mary’s Hospital, Seoul 06591, Republic of Korea; hakjjang@catholic.ac.kr

**Keywords:** ampulla of Vater cancer, EGFR, HER2, c-Met, prognosis, recurrence

## Abstract

**Simple Summary:**

Adenocarcinoma of the ampulla of Vater (AAC) is a rare cancer that arises from various histologic subtypes, creating a diverse group of tumors in need of new therapeutic strategies. This study examines the expression of epidermal growth factor receptor (EGFR), human epidermal growth factor receptor 2 (HER2), and c-Met in AAC to explore their potential as druggable targets. Among 87 patients, EGFR overexpression was found in 87.4%, HER2 in 11.5%, and c-Met in 50%. EGFR overexpression was more common in the pancreatobiliary subtype and linked to a higher histologic grade. EGFR-positive AAC patients had worse disease-free and overall survival compared to EGFR-negative patients. Although HER2-positive AAC showed a trend towards shorter survival, it was not statistically significant. In systemic disease, survival outcomes were unaffected by EGFR, HER2, and c-Met expression, although the HER2-positive group tended towards worse progression-free survival. These results highlight the need for further research to evaluate targeted treatments in AAC patients with these protein expressions.

**Abstract:**

Adenocarcinoma of the ampulla of Vater (AAC) is a rare malignancy with heterogeneous tumors arising from various histologic subtypes, necessitating new therapeutic strategies. This study examines epidermal growth factor receptor (EGFR), human epidermal growth factor receptor 2 (HER2), and c-Met expression in AAC, given their potential as druggable targets. Among 87 patients who underwent curative resection, EGFR overexpression was found in 87.4%, HER2 in 11.5%, and c-Met in 50%. EGFR overexpression was more common in the pancreatobiliary subtype (*p* = 0.018) and associated with a higher histologic grade (*p* = 0.008). HER2 did not correlate with clinicopathological features, while c-Met was more common in node-negative groups (*p* = 0.004) and often co-expressed with EGFR (*p* = 0.049). EGFR-positive patients had worse disease-free (HR = 2.89; 95% CI, 1.35–6.20; *p* = 0.061) and overall survival (HR = 6.89; 95% CI, 2.94–16.2; *p* = 0.026) than EGFR-negative patients. HER2-positive AAC showed a trend towards shorter survival, although not statistically significant, and c-Met had no impact on survival outcomes. In the context of systemic disease, survival outcomes did not vary according to EGFR, HER2, and c-Met expression, but the HER2-positive group showed a trend towards inferior progression-free survival (HR = 1.90; 95% CI, 0.56–6.41; *p* = 0.166). This study underscores the potential of EGFR, HER2, and c-Met as targets for personalized therapy in AAC, warranting further research to evaluate targeted treatments.

## 1. Introduction

Adenocarcinoma of the ampulla of Vater (AAC) is a rare malignancy, comprising approximately 7% of all periampullary cancers [1]. Surgical resection is the primary curative treatment for patients with localized AAC. However, the recurrence rate remains high, and the 5-year survival rate is reported to be only 45% [2]. For advanced AAC, there is limited data to guide physicians in selecting chemotherapy treatments. Despite the lack of robust evidence, gemcitabine-based regimens have frequently been used, as supported by the ABC-02 trial [3]. Additionally, the combination of capecitabine and oxaliplatin has been studied as an effective first-line therapy for patients with advanced AAC [4]. 

Based on histological features and specific immunohistochemical markers, AACs have been classified into two histologic subtypes: pancreatobiliary (PB) type and intestinal type [5,6]. As of 2010, the World Health Organization (WHO) revised this classification by adding a mixed subtype, resulting in three distinct AAC subtypes [7]. The PB subtype is generally more aggressive and linked to poorer clinical outcomes compared to the intestinal subtype [8]. Given these differences, physicians may tailor chemotherapy regimens according to the histologic subtype [9]. Other than these therapeutic strategies, developing effective targeted agents combined with chemotherapy for AACs is challenging due to the rarity of the disease and the limited understanding of druggable molecular alterations.

As one of the druggable molecules, the epidermal growth factor receptor (EGFR) is activated by its ligands, leading to receptor dimerization and signal propagation to downstream pathways. This EGFR-mediated activation triggers oncogenic signaling cascades, including the mitogen-activated protein kinase (MAPK) and phosphatidylinositol 3-kinase (PI3K) pathways, which promote tumor growth and progression [10]. EGFR activation has been implicated in the development of many solid tumors, such as colorectal cancer and pancreatic ductal adenocarcinoma [11,12]. In patients with AAC, EGFR overexpression has been observed in the PB subtype, but its relationship to the prognosis has not been clearly established [13]. 

Another well-known druggable molecule is the human epidermal growth factor receptor 2 (HER2), which has been implicated in the pathogenesis of various human cancers [14]. HER2 overexpression has been associated with a poor prognosis in breast cancer, gastric cancer, and biliary tract cancer [15,16,17]. Clinical trials have shown that HER2-targeted therapies improve clinical outcomes in patients with HER2-positive breast and gastric cancers [18,19]. Although previous studies have reported HER2 overexpression in 13–16% of patients with AAC, the clinical characteristics and prognostic implications of HER2 overexpression in AAC remain unclear [20,21]. 

c-Met, a receptor tyrosine kinase, acts as a high-affinity receptor for hepatocyte growth factor (HGF). Activation of the HGF/c-Met pathway facilitates cell invasiveness and metastasis by promoting cell proliferation, invasion, resistance to apoptosis, and angiogenesis [22]. c-Met is upregulated in various cancers, including biliary tract and pancreatic cancers, and is correlated with an unfavorable prognosis [23,24]. In patients with periampullary cancer, c-Met overexpression combined with *KRAS* mutations significantly increases the risk of recurrence in the PB subtype of AAC [25]. 

This study evaluated the expression of potentially therapeutic targetable proteins, EGFR, HER2, and c-Met, using immunohistochemistry (IHC) labeling on tissue microarray (TMA) samples from surgically resected AAC. We investigated the relationship between EGFR, HER2, and c-Met expression and the clinicopathological features and prognostic value in patients with AAC.

## 2. Materials and Methods

### 2.1. Patients and Data Acquisition

In this study, we analyzed a cohort of patients with surgically resected adenocarcinoma of the AAC that has been previously reported in a published study [26]. The current analysis focuses on evaluating the expression of proteins such as EGFR, HER2, and c-Met in the same patient cohort. The inclusion and exclusion criteria for this study were identical to those in the previously published study. Patients eligible for inclusion were required to meet the following criteria: (1) histologically confirmed adenocarcinoma of the ampulla of Vater; (2) pathologic stage IB–III as per the 8th edition of the American Joint Committee on Cancer Staging; and (3) verifiable recurrence and survival data at the time of collection. Exclusion criteria were: (1) pathological tumor, node, metastasis (TNM) stage IA disease; (2) absence of examined regional lymph nodes; (3) presence of macroscopically residual tumors (R2 resection); (4) receipt of preoperative chemotherapy or radiotherapy; or (5) diagnosis of a secondary malignancy post-surgery.

### 2.2. Tissue Microarray and Immunohistochemistry

Formalin-fixed paraffin-embedded (FFPE) tissue specimens from patients with AAC were obtained from the pathological archives. Two expert pancreatic histopathologists (SHL and YK), blinded to clinical outcomes, independently reviewed and confirmed all the pathological features. For each sample, a pathologist (YK) annotated at least two areas of the tumor. Using these annotations, two 2 mm diameter core tissues were extracted and rearranged into a new recipient TMA block with a trephine apparatus. The FFPE TMA sections were deparaffinized with xylene and subsequently rehydrated. Antigen retrieval and immunostaining were performed using an automated Bond-max immunostainer (Leica Microsystems, Newcastle, UK). Primary antibodies for anti-EGFR (clone 3C6), anti-HER2/neu (clone 4B5; dilution: 1:100), and anti-c-Met (clone SP44; all from Ventana, Tucson, AZ, USA), as well as monoclonal antibodies for anti-MUC1 (Novocastra, Newcastle, UK), anti-MUC2 (Novocastra, Newcastle, UK), anti-CDX2 (BioGenex, CA, USA), and anti-CK20 (Santa Cruz Biotechnology, TX, USA), were applied. Antibody binding was detected using the Bond Polymer Refine Detection Kit (catalog #DS9800; Leica Microsystems, Vista, CA, USA).

### 2.3. Histologic Subtypes

The tumors were classified into two histologic subtypes based on morphology: the intestinal subtype, characterized by tall columnar cells forming elongated glands similar to colorectal adenocarcinoma, and the PB subtype, defined by cells with rounded nuclei forming rounded glands typical of pancreatobiliary carcinomas. Mixed type classification was applied if both subtype characteristics were present. Four immunohistochemical markers (MUC1, CDX2, CK20, and MUC2) were evaluated using IHC labeling on the TMA samples. Initial categorization was based on immunohistochemical staining, with ambiguous cases relying on morphological evaluation and CDX2 and MUC1 expression criteria [6]. 

### 2.4. Immunohistochemical Analysis

The TMA slides were scanned with an Aperio AT2 (Leica Biosystems, Wetzlar, Germany). The membranous staining of EGFR, HER2, and c-Met was evaluated on a scale from 0 (no staining) and 1+ (faint staining) through 2+ (weak to moderate staining) to 3+ (strong staining). In cases of heterogeneity, the most intense staining that covered more than 10% of the tumor cells was chosen as the representative score. Staining for HER2 was classified as negative (scores of 0 and 1+), equivocal (2+), or positive (3+), whereas staining for EGFR and c-Met was classified as either negative (0) or positive (any level of membranous staining greater than 10%, including scores of 1+, 2+, or 3+) (Appendix A). When there was a discrepancy in diagnosis, two pathologists discussed it to reach an agreement.

### 2.5. Treatment and Surveillance

Pancreaticoduodenectomy or pylorus-preserving pancreaticoduodenectomy with standard lymph node dissection was performed at the surgeon’s discretion. The decision regarding adjuvant chemotherapy and the selection of the chemotherapy regimen were made by the treating physician. Fluorouracil-based adjuvant chemotherapy was initiated within 12 weeks after surgery. Adjustments to the chemotherapy dosage and schedule were allowed as necessary. Postoperative assessments were conducted every 3 months for the first 2 years, every 6 months for the next 3 years, and annually thereafter. Imaging evaluations utilized computed tomography, and levels of carbohydrate antigen 19-9 (CA 19-9) were measured at each visit.

### 2.6. Statistical Analysis

The statistics are presented as proportions or medians with ranges. Chi-square or Fisher’s exact test compared the categorical variables, and an unpaired *t*-test compared the continuous variables. Disease-free survival (DFS) was defined as the time between curative surgery and recurrence or death from any cause. Overall survival (OS) was determined from the date of surgery to the last follow-up or death from any cause. Survival outcomes were estimated using Kaplan–Meier and compared with the log-rank test. The prognostic impact of clinicopathological factors on DFS and OS was assessed using the Cox proportional hazards model. For multivariate analysis, Cox regression was performed using a forward stepwise approach, including variables significant in univariate analysis (*p* < 0.05) as well as clinically relevant variables known from the previous literature to be associated with survival outcomes. All tests were two-sided, and a *p*-value < 0.05 was considered statistically significant. Statistical analyses were carried out using SPSS for Windows version 24.0 (IBM SPSS Inc., Armonk, NY, USA) and GraphPad Prism version 10.2.3 (GraphPad Software Inc., San Diego, CA, USA).

## 3. Results

### 3.1. Patients Characteristics

From 1 August 2007 to 31 December 2021, a total of 87 patients were included in this study. The demographic and clinicopathological characteristics of these patients, previously reported in an earlier study [26], are summarized in Table 1. These patients with AAC were categorized into PB type (n = 54) and intestinal type (n = 33). Both subtypes had similar median ages, gender distributions, mean tumor sizes, and histologic grading. The intestinal subtype had a higher proportion of T1-2 tumors compared to the PB subtype (66.7% vs. 46.3%; *p* = 0.064), although node category and TNM stage distributions were similar (*p* = 0.380 and *p* = 0.480, respectively). EGFR expression was significantly higher in the PB type (94.4% vs. 75.8%; *p* = 0.018), whereas HER2 and c-Met expressions did not show significant differences (*p* = 0.571 and *p* = 0.467, respectively). 

### 3.2. Characteristics of Patients Stratified by EGFR, HER2, and c-Met Expression

The analysis of clinicopathological factors based on the expression of EGFR, HER2, and c-Met revealed several significant differences (Appendix A). EGFR expression was significantly associated with higher histologic grading, with 86.8% of EGFR-positive tumors classified as Grade 2/3 (*p* = 0.008). Furthermore, EGFR positivity frequently co-occurred with c-Met positivity (*p* = 0.049). c-Met expression demonstrated a significant correlation with both node category and TNM stage, with a higher proportion of c-Met positive patients being classified as N0 (58.1% vs. 41.9%; *p* = 0.004) and Stage I–II (60.5% vs. 39.5%; *p* = 0.002). Additionally, c-Met positivity was significantly associated with the absence of lymphatic invasion (53.5% vs. 46.5%; *p* = 0.041).

### 3.3. Survival Outcomes

The median follow-up period for the patient cohort was 32.8 months (95% CI, 25.3–49.2). During this time, 52 patients (59.8%) experienced recurrence, and 46 patients (52.9%) died. The median DFS for the entire group was 19.3 months (95% CI, 0.0–37.2), with 1-year and 2-year DFS rates of 64.7% (95% CI, 53.5–73.8%) and 47.9% (95% CI, 36.9–58.1%), respectively. The median OS was 59.4 months (95% CI, 42.8–76.0), with estimated OS rates of 77.0% (95% CI, 66.7–84.5%) at 2 years and 49.5% (95% CI, 37.6–60.3%) at 5 years.

### 3.4. Survival Analyses Based on EGFR Expression

Survival analysis based on EGFR expression revealed recurrence in 49 (64.5%) of 76 patients with positive EGFR expression, while it occurred in 3 (27.3%) of 11 patients with negative EGFR expression. The median DFS was 16.9 months (95% CI, 10.5–23.3) for the EGFR-positive group, while it was not reached for the EGFR-negative group (HR = 2.89; 95% CI, 1.35–6.20; *p* = 0.061; Figure 1A). Deaths occurred in 45 (59.2%) of the 76 EGFR-positive patients and 1 (9.1%) of the 11 EGFR-negative patients. The median OS was 53.6 months (95% CI, 36.7–70.6) for the EGFR-positive group, compared to not reached for the EGFR-negative group (HR = 6.89; 95% CI, 2.94–16.2; *p* = 0.026; Figure 1B). The 5-year OS rates were 45.4% (95% CI, 33.3–56.7%) for EGFR-positive patients and 90.9% (95% CI, 50.8–98.7%) for EGFR-negative patients.

### 3.5. Survival Outcomes in Relation to HER2 and c-Met Expression

Survival analysis based on HER2 expression revealed a median DFS of 15.1 months (95% CI, 0.0–29.0) for HER2-positive patients, compared to 27.3 months (95% CI, 0.0–39.9) for HER2-negative patients (HR = 1.78; *p* = 0.128; Figure 1C), indicating no significant difference. Similarly, the median OS was 37.8 months (95% CI, 10.4–65.1) for the HER2-positive group, while it was 65.3 months (95% CI, 41.8–88.9) for the HER2-negative group (HR = 1.68; *p* = 0.196; Figure 1D), also showing no significant difference. In terms of c-Met expression, the median DFS was 22.2 months (95% CI, 0.0–50.0) for c-Met-positive patients, whereas it was 17.2 months (95% CI, 0.0–40.1) for c-Met-negative patients (HR = 0.94; *p* = 0.822; Figure 1E), again with no statistical difference. The median OS for c-Met-positive patients was 60.3 months (95% CI, 34.1–86.5), compared to 59.4 months (95% CI, 43.0–75.9) for c-Met-negative patients (HR = 0.92; *p* = 0.763; Figure 1F), also showing no significant difference.

### 3.6. Multivariate Analysis of Survival Outcomes 

Multivariable analysis revealed that an advanced tumor stage (HR = 2.20; 95% CI, 1.23–3.92; *p* = 0.008) correlated with a shorter DFS (Table 2). Positive EGFR expression, though trending toward shorter DFS compared to negative expression, did not reach statistical significance (HR = 2.38; 95% CI, 0.69–8.23; *p* = 0.172). Assessing OS through multivariate analysis showed a significant association between advanced tumor stage and poorer survival outcomes (HR = 2.44; 95% CI, 1.30–4.58; *p* = 0.006; Table 3). Patients with EGFR overexpression, while not statistically significant, tended to exhibit inferior OS compared to those with negative expression (HR = 6.26; 95% CI, 0.81–48.4; *p* = 0.079).

### 3.7. Survival Outcomes after Recurrence

Recurrence was identified in 52 patients, with 38 (73.1%) of these individuals undergoing palliative systemic chemotherapy. Disease progression was subsequently confirmed in 34 of these 38 patients (89.5%), which formed the basis for calculating progression-free survival (PFS). From the start of first-line systemic chemotherapy, the median follow-up period was 14.9 months (95% CI, 8.95–22.6). Differences in PFS based on EGFR, HER2, and c-Met expression were not statistically significant (Figure 2). The HER2-positive group had a median PFS of 4.4 months (95% CI, 4.08–4.64), compared to 6.0 months (95% CI, 5.05–7.02) in the HER2-negative group, showing a trend towards inferior PFS in the HER2-positive group, though not statistically significant (HR = 1.90; 95% CI, 0.56–6.41; *p* = 0.166; Figure 2C). Of the 38 patients who underwent palliative chemotherapy, 33 (86.8%) died, forming the basis for OS analysis from the start of systemic treatment. Survival outcomes did not differ according to EGFR, HER2, and c-Met expression in patients treated with palliative systemic chemotherapy (Figure 2).

## 4. Discussion

In this analysis of surgically resected stage IB–III AAC patients, positive EGFR expression was more frequently observed in the PB subtype and was associated with a higher histologic grade. Additionally, EGFR-positive AAC patients exhibited inferior DFS and OS outcomes compared to those with negative EGFR expression. While the expressions of EGFR, HER2, and c-Met did not show significant correlations with PFS and OS in patients who received palliative chemotherapy following recurrence, HER2-positive AAC patients demonstrated a trend towards worse PFS. This pioneering study is the first to examine recurrence and survival outcomes in relation to druggable alterations (EGFR, HER2, and c-Met overexpression) among surgically resected ampullary adenocarcinoma patients at risk of recurrence, excluding very early-stage cases.

EGFR overexpression was identified in the majority (87.4%) of AAC patients, which represents a higher proportion compared to the approximately 25–75% observed in colorectal cancers and about 45% in pancreatic ductal adenocarcinoma [27,28]. Due to the absence of a standardized definition for EGFR positive expression through IHC in AAC patients, it is necessary to establish criteria for positive expression by selecting patients who exhibit strong intensity. A significant majority (94.4%) of the PB subtype demonstrated EGFR expression, aligning with previous research findings that EGFR expression is more frequently observed in the PB type than in the intestinal type among AAC patients [13]. Furthermore, EGFR-positive expression was associated with moderate to poorly differentiated histology. These clinical factors associated with EGFR suggest that EGFR-positive AAC patients have poorer DFS and OS outcomes compared to EGFR-negative patients. This is consistent with findings in colorectal and pancreatic cancers, where EGFR overexpression has been linked to a poor prognosis and an increased risk of metastasis [11,27]. However, while EGFR-positive patients exhibited a trend towards inferior DFS, this trend did not reach statistical significance. This lack of statistical significance may be attributed to the limited sample size, with a majority (87.4%) of patients classified as EGFR-positive. Additionally, a considerable proportion (12 of 52, 23.1%) of recurrent cases experienced local recurrence exclusively, indicating that the advanced tumor stage may have exerted the most substantial influence on disease recurrence. In patients who received systemic chemotherapy after recurrence, no prognostic differences based on EGFR expression were observed. This likely results from most EGFR-negative patients not experiencing recurrence and thus not receiving systemic treatment. Validation with a larger sample size or redefining EGFR-positive expression criteria is needed for further analysis.

In our study, HER2 overexpression was identified in 11.5% of patients, which is consistent with the incidence of HER2 overexpression previously reported in AAC patients [20,21]. The HER2-positive AAC group did not show any associations with other clinicopathological factors. Although not statistically significant, HER2-positive AAC exhibited numerically shorter DFS and OS compared to HER2-negative AAC, and a trend towards shorter PFS in patients who received systemic treatment after recurrence. These findings align with previous research indicating that HER2 overexpression in biliary tract cancer correlates with reduced DFS, OS, and PFS in patients receiving palliative gemcitabine with cisplatin treatment [28]. Furthermore, HER2 overexpression is recognized as a negative prognostic factor and a negative predictive biomarker for anti-EGFR treatment in metastatic colorectal cancer, reinforcing its role in poor outcomes across different histologic types [29]. Although AAC patients with HER2 overexpression did not show a significant association with survival outcomes, possibly because of their small numbers, the potential utility of HER2-targeted therapeutic strategies in this subpopulation warrants further investigation.

c-Met positive expression was identified in approximately half of the AAC patients and was more frequently observed in the node-negative, lower TNM stage group. Nonetheless, no differences in DFS and OS were observed between the two groups stratified by c-Met expression. These findings align with a previous study indicating that c-Met overexpression in the PB subtype of AAC is not associated with survival outcomes [25]. The lack of correlation between lower pathological stage and favorable survival outcomes could be due to several reasons. In our multivariate analysis of DFS and OS, node involvement did not show a significant association with survival outcomes. Thus, even though c-Met positive expression was more frequent in the node-negative group, it may not have impacted survival outcomes. Additionally, the significant co-expression with EGFR overexpression, a known negative prognostic factor, might have influenced the prognosis. According to a previous study, the co-expression of c-Met and EGFR is significantly correlated in both intrahepatic and extrahepatic cholangiocarcinoma and is associated with poor prognosis [23]. However, the association between c-Met overexpression and poor prognosis is more prominent in intrahepatic cholangiocarcinoma and not significant in extrahepatic cholangiocarcinoma. Given that AAC anatomically resembles extrahepatic cholangiocarcinoma, our findings are consistent with these previous results.

Overexpression of EGFR, HER2, and c-Met represents significant molecular alterations in various tumors, serving as crucial targets for treatment strategies. Anti-EGFR monoclonal antibodies, when combined with chemotherapy, demonstrate a synergistic effect in advanced colorectal cancer and are part of the standard treatment regimen [30]. However, patients with colorectal tumors harboring *KRAS* mutations do not benefit from anti-EGFR treatment [31]. Previous studies indicate that approximately 40–65% of AAC patients harbor *KRAS* mutations [8], suggesting that around 30–50% could benefit from combining anti-EGFR therapy with chemotherapy. HER2-directed therapies, including anti-HER2 monoclonal antibodies (trastuzumab, pertuzumab), tyrosine kinase inhibitors (lapatinib, neratinib, tucatinib), and HER2-directed antibody–drug conjugates (ado-trastuzumab emtansine, fam-trastuzumab deruxtecan), have demonstrated efficacy across multiple cancers [18]. Notably, recent evidence has shown that various anti-HER2 treatments are effective in advanced biliary tract cancer patients with HER2 overexpression, who otherwise have limited treatment options [32,33]. Therefore, AAC patients with HER2 overexpression may also benefit from these therapies. Although the prognostic significance of c-Met overexpression in AAC remains uncertain, its frequent co-expression with EGFR, which is associated with a poor prognosis, suggests that novel targeted treatments such as amivantamab, a bispecific antibody targeting EGFR and c-Met, may be efficacious [34].

This study has several limitations. First, protein expression was verified solely through immunohistochemical analysis, without additional validation via in situ hybridization (ISH) or next-generation sequencing (NGS) for gene amplification. Consequently, there may be potential discrepancies between protein overexpression and gene amplification. Moreover, as expression intensity for EGFR and c-Met was not scored, further discussion is needed to refine the definition of positive expression based on intensity. Nonetheless, EGFR protein expression via IHC has been strongly associated with gene amplification in lung adenocarcinoma and breast cancer when detected by fluorescence and chromogenic ISH, respectively [35,36]. Similarly, in gastric cancer, MET mRNA expression through RNA ISH aligns well with c-Met protein expression by IHC [37]. However, in lung cancer, c-Met IHC shows a poor correlation with fluorescence ISH and NGS results [38]. This indicates that the reliability of c-Met IHC may vary by cancer type, highlighting the need for complementary molecular validation in future research. Second, a comprehensive analysis of genetic alterations that could confer resistance to anti-EGFR and anti-HER2 targeted treatments, such as *KRAS* mutations, was not conducted. Further research is necessary to closely examine the relationship between these molecular alterations and prognosis. Lastly, the analysis relied solely on surgical tissue samples, without additional biopsies for recurrent patients. This limitation could introduce tumor heterogeneity as a confounding factor in the survival outcomes of these patients.

## 5. Conclusions

This study investigated the association between EGFR, HER2, and c-Met expression and clinicopathological characteristics and survival outcomes in surgically treated patients with AAC. The overexpression of EGFR, HER2, and c-Met represents potentially druggable alterations, necessitating further research to evaluate the efficacy of targeted therapies in AAC patients exhibiting these protein expressions.

## Figures and Tables

**Figure 1 cancers-16-02756-f001:**
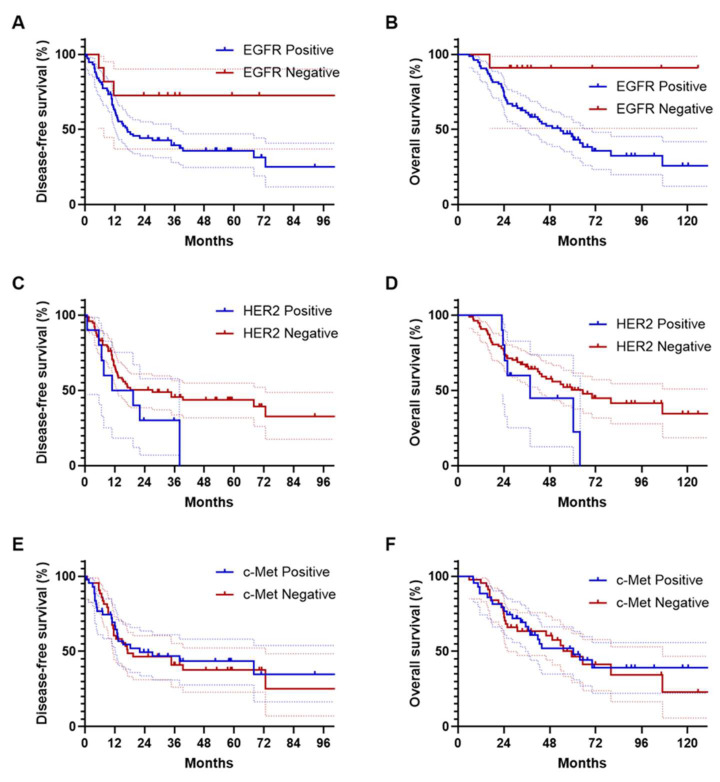
Kaplan–Meier estimates of disease-free survival (**A**,**C**,**E**) and overall survival (**B**,**D**,**F**) in patients with ampulla of Vater cancer, stratified by EGFR, HER2, and c-Met expression.

**Figure 2 cancers-16-02756-f002:**
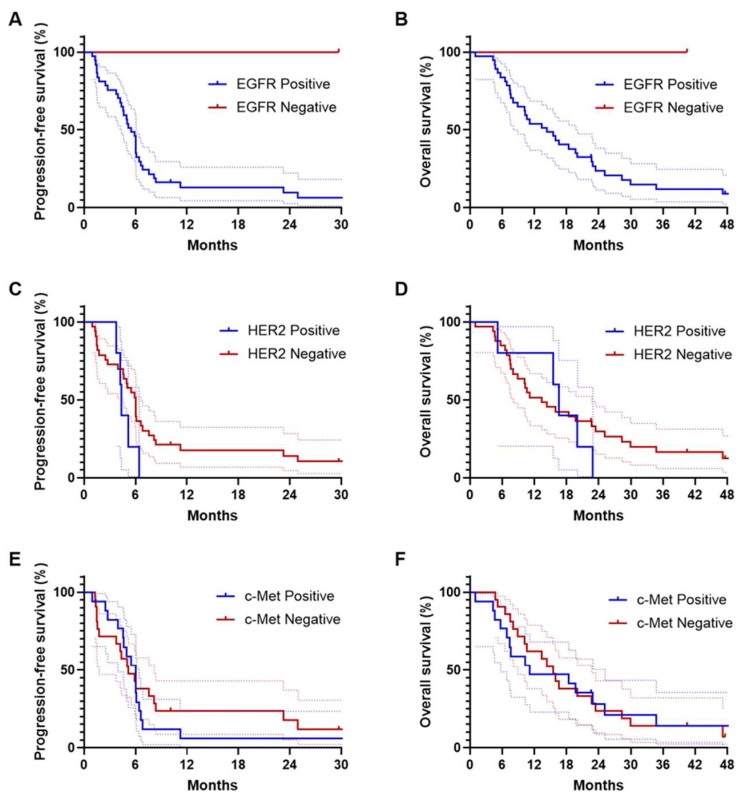
Kaplan–Meier estimates of progression-free survival (**A**,**C**,**E**) and overall survival (**B**,**D**,**F**), according to EGFR, HER2, and c-Met expression.

**Table 1 cancers-16-02756-t001:** Demographic and clinicopathological features of patients with surgically treated adenocarcinoma of the ampulla of Vater.

Variables	Total (n = 87)	PB Type (n = 54)	Intestinal Type (n = 33)	*p* Value
Age, Median (range)	65 (39–87)	66 (39–87)	65 (42–87)	0.789
Gender, n (%)				
Male	49 (56.3)	29 (53.7)	20 (60.6)	0.529
Female	38 (43.7)	25 (46.3)	13 (39.4)	
Tumor size, cm, mean ± SD	2.5 ± 1.1	2.4 ± 1.0	2.5 ± 1.1	0.691
Histologic grading, n (%)				
Grade 1	15 (17.2)	7 (13.0)	8 (24.2)	0.177
Grade 2/3	72 (82.8)	47 (87.0)	25 (75.8)	
Tumor Category, n (%)				
T1–2	47 (54.0)	25 (46.3)	22 (66.7)	0.064
T3–4	40 (46.0)	29 (53.7)	11 (33.3)	
Node Category, n (%)				
N0	37 (42.5)	21 (38.9)	16 (48.5)	0.380
N1–2	50 (57.5)	33 (61.1)	17 (51.5)	
TNM Stage *, n (%)				
Stage I–II	38 (43.7)	22 (40.7)	16 (48.5)	0.480
Stage III–IV	49 (56.3)	32 (59.3)	17 (51.5)	
EGFR expression, n (%)				
Positive	76 (87.4)	51 (94.4)	25 (75.8)	0.018
Negative	11 (12.6)	3 (5.6)	8 (24.2)	
HER2 expression, n (%)				
Positive	10 (11.5)	6 (11.1)	4 (12.1)	0.571
Negative	77 (88.5)	48 (88.9)	29 (87.9)	
c-Met expression, n (%)				
Positive	43 (49.4)	26 (48.1)	17 (51.5)	0.467
Negative	44 (50.6)	28 (51.9)	16 (48.5)	
Adjuvant chemotherapy, n (%)				
No	44 (50.6)	26 (48.1)	18 (54.5)	0.563
Yes	43 (49.4)	28 (51.9)	15 (45.5)	
Preoperative CA19-9 level, n (%)				
Within normal (<40 U/mL)	39 (44.8)	20 (37.0)	19 (57.6)	0.151
Above normal (≥40 U/mL)	40 (46.0)	29 (53.7)	11 (33.3)	
Missing data	8 (9.2)	5 (9.3)	3 (9.1)	

PB, pancreatobiliary; SD, standard deviation; TNM, tumor node metastasis; EGFR, epidermal growth factor receptor; HER2, human epidermal growth factor receptor 2; CA19-9 carbohydrate antigen 19-9. * According to the American Joint Committee on Cancer 8th edition.

**Table 2 cancers-16-02756-t002:** Univariate and multivariate analyses of clinicopathologic features affecting disease-free survival in patients with ampulla of Vater adenocarcinoma.

	DFS			
Variables	Univariate Analysis		Multivariate Analysis	
	HR (95% CI)	*p* Value	HR (95% CI)	*p* Value
Age ≥ 70 (vs. <70 year)	1.33 (0.76–2.32)	0.319		
Histologic grade 2–3 (vs. grade 1)	2.60 (1.03–6.55)	0.043	1.94 (0.75–4.99)	0.171
**Tumor stage 3 or 4 (vs. Stage 1 or 2)**	**2.49 (1.43–4.34)**	**0.001**	**2.20 (1.23–3.92)**	**0.008**
Nodal metastasis (vs. none)	2.01 (1.12–3.61)	0.019	1.13 (0.61–2.09)	0.702
Pancreatobiliary type (vs. intestinal)	1.74 (0.95–3.17)	0.072	1.20 (0.62–2.32)	0.588
EGFR expression (vs. none)	2.89 (1.35–6.20)	0.061	2.38 (0.69–8.23)	0.172
Received AC (vs. none)	1.04 (0.60–1.79)	0.899		

DFS, disease-free survival; HR, hazard ratio; EGFR, epidermal growth factor receptor; AC, adjuvant chemotherapy. Statistically significant variables are in bold font.

**Table 3 cancers-16-02756-t003:** Univariate and multivariate assessment of clinicopathologic factors in relation to overall survival in adenocarcinoma of the ampulla of Vater.

	OS			
Variables	Univariate Analysis		Multivariate Analysis	
	HR (95% CI)	*p* Value	HR (95% CI)	*p* Value
Age ≥ 70 (vs. <70 year)	1.75 (0.96–3.18)	0.069	1.74 (0.95–3.20)	0.074
Histologic grade 2–3 (vs. grade 1)	1.98 (0.70–5.55)	0.197		
**Tumor stage 3 or 4 (vs. Stage 1 or 2)**	**2.60 (1.42–4.76)**	**0.002**	**2.44 (1.30–4.58)**	**0.006**
Nodal metastasis (vs. none)	1.70 (0.92–3.14)	0.088	1.46 (0.77–2.75)	0.243
Pancreatobiliary type (vs. intestinal)	2.09 (1.06–4.14)	0.034	1.15 (0.56–2.39)	0.706
EGFR expression (vs. none)	6.89 (2.94–16.2)	0.026	6.26 (0.81–48.4)	0.079
Received AC (vs. none)	0.96 (0.53–1.72)	0.877		

OS, overall survival; HR, hazard ratio; EGFR, epidermal growth factor receptor; AC, adjuvant chemotherapy. Statistically significant variables are in bold font.

## Data Availability

The datasets used in the current study are available from the corresponding author on request.

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
