# Peer review of "Prognostic Significance of EGFR, HER2, and c-Met Overexpression in Surgically Treated Patients with Adenocarcinoma of the Ampulla of Vater"

_cancers, 2024, doi:10.3390/cancers16152756_

Round 1
Reviewer 1 Report
Comments and Suggestions for Authors
The study sought to determine the prognostic significance of EGFR, HER2 and c-Met over expression using immunohistochemistry in Adenocarcinoma of the Ampulla of Vater(AAC) on 87 surgically resected patients tissue microarrays. It showed that there was a significant relationship between EGFR overexpression and worse disease-free survival (DFS) and overall survival (OS). This study also highlights the possibilities of targeting these markers for personalized therapy in AAC.
1- The reason why EGFR, HER2, and c-Met were chosen as the focus is not very clear in this article. Explain further why these particular markers were selected giving details about their known role in other cancers as well as preliminary data illustrating their relevance to AAC.
2-While the article presents univariate and multivariate analyses, the selection criteria for variables included in multivariate models need to be clarified.
3-Some p values ​​approach but do not reach statistical significance (e.g., EGFR and DFS). Discussing these trends in more detail can add depth to the analysis.
4- Criteria for positive EGFR and c-Met expression are not clearly defined. Consider providing more detailed scoring methods and discussing potential variability.
5- Assessment of protein expression using IHC alone is controversial due to potential variability in interpretation and lack of quantitative precision. Consider discussing the limitations of using IHC alone and recommend complementary techniques (e.g., in situ hybridization, next-generation sequencing) for confirmation.
6- The English language used throughout the manuscript needs some improvements in terms of style and grammar ( for example-Table 1 Missig data )
Comments on the Quality of English LanguageModerate editing of English language required
Author Response
Reviewer Comments:
Reviewer 1
The study sought to determine the prognostic significance of EGFR, HER2 and c-Met over expression using immunohistochemistry in Adenocarcinoma of the Ampulla of Vater (AAC) on 87 surgically resected patients tissue microarrays. It showed that there was a significant relationship between EGFR overexpression and worse disease-free survival (DFS) and overall survival (OS). This study also highlights the possibilities of targeting these markers for personalized therapy in AAC.
Comment 1: The reason why EGFR, HER2, and c-Met were chosen as the focus is not very clear in this article. Explain further why these particular markers were selected giving details about their known role in other cancers as well as preliminary data illustrating their relevance to AAC.
Response 1: Thank you for your insightful comments. It appears that the purpose of this study was not clearly articulated in the Introduction. EGFR, HER2, and c-Met are among the therapeutic biomarkers for which targeted treatments are most advanced. Specifically, anti-EGFR and anti-HER2 directed treatments have been used for many years in other solid tumors. Recent studies have shown their efficacy in a pan-tumor context, suggesting the possibility of tumor-agnostic indications. Given that treatment options for AAC are highly limited, this study aimed to investigate the relevance of these biomarkers, which already have established targeted treatments. Accordingly, we have revised the Introduction section to better elucidate the study's purpose, incorporating data from other tumors and preliminary data from AAC, as per your suggestion.
The manuscript is revised as follows:
Based on histological features and specific immunohistochemical markers, AACs have been classified into two histologic subtypes: pancreato-biliary (PB) type and intestinal type [5,6]. As of 2010, the World Health Organization (WHO) revised this classification by adding a mixed subtype, resulting in three distinct AAC subtypes [7]. The PB subtype is generally more aggressive and linked to poorer clinical outcomes compared to the intestinal subtype [8]. Given these differences, physicians may tailor chemotherapy regimens according to the histologic subtype [9]. Other than these therapeutic strategies, developing effective targeted agents combined with chemotherapy for AACs is challenging due to the rarity of the disease and the limited understanding of druggable molecular alterations.
As one of the druggable molecules, the epidermal growth factor receptor (EGFR) is activated by its ligands, leading to receptor dimerization and signal propagation to downstream pathways. This EGFR-mediated activation triggers oncogenic signaling cascades, including the mitogen-activated protein kinase (MAPK) and phosphatidylinositol 3-kinase (PI3K) pathways, which promote tumor growth and progression [10]. EGFR activation has been implicated in the development of many solid tumors, such as colorectal cancer and pancreatic ductal adenocarcinoma [11,12].
Another well-known druggable molecule is the Human Epidermal Growth Factor Receptor 2 (HER2), which has been implicated in the pathogenesis of various human cancers [14]. HER2 overexpression has been associated with poor prognosis in breast cancer, gastric cancer, and biliary tract cancer [15-17]. Clinical trials have shown that HER2-targeted therapies improve clinical outcomes in patients with HER2-positive breast and gastric cancers [18,19].
This study evaluated the expression of potentially therapeutic targetable proteins, EGFR, HER2, and c-Met, using immunohistochemistry (IHC) labeling on tissue micro-array (TMA) samples from surgically resected AAC. We investigated the relationship between EGFR, HER2, and c-Met expression and the clinicopathological features and prognostic value in patients with AAC.
Comment 2: While the article presents univariate and multivariate analyses, the selection criteria for variables included in multivariate models need to be clarified.
Response 2: We sincerely appreciate your supportive feedback and insightful comments.
Our variable selection process for the multivariate analysis was twofold: We included variables that demonstrated significant associations with survival outcomes in the univariate analysis. Additionally, we incorporated variables that, while not showing significant associations in the univariate analysis, are well-established in the literature as having significant correlations with survival outcomes. Examples of such variables include advanced age and nodal involvement.
We recognize the critical importance of variable selection in determining statistical outcomes. To ensure transparency, we have provided a detailed description of our selection methodology in the methods section of our manuscript. We acknowledge a limitation of our study: as a retrospective analysis with a relatively small sample size, the inclusion of multiple variables may potentially diminish the statistical significance of our results. We are grateful for the opportunity to clarify these points and hope this explanation addresses your concerns. Your feedback has been invaluable in helping us refine and improve our work.
The manuscript is revised as follows:
Overall survival (OS) was determined from the date of surgery to the last follow-up or death from any cause. Survival outcomes were estimated using Kaplan-Meier and compared with the log-rank test. The prognostic impact of clinicopathological factors on DFS and OS was assessed using the Cox proportional hazards model. For multivariate analysis, Cox regression was performed using a forward stepwise approach, including variables significant in univariate analysis (p < .05) as well as clinically relevant variables known from previous literature to be associated with survival outcomes.
Comment 3: Some p values approach but do not reach statistical significance (e.g., EGFR and DFS). Discussing these trends in more detail can add depth to the analysis.
Response 3: We sincerely appreciate the reviewer's perceptive observations. We believe that the lack of statistical significance, despite the observed trends towards inferior survival outcomes, is primarily due to the small sample size. This is a common challenge when studying rare diseases like AAC, where obtaining a large sample size is difficult. Nonetheless, comparing our findings with studies on similar tumors, such as colorectal cancer and biliary tract cancer, suggests that EGFR and HER2 overexpression may have prognostic value and potential therapeutic implications. Based on the reviewer's comments, we have revised the discussion section to reflect these points.
The manuscript is revised as follows:
Furthermore, EGFR positive expression was associated with moderate to poorly differentiated histology. These clinical factors associated with EGFR suggest that EGFR positive AAC patients have poorer DFS and OS outcomes compared to EGFR negative patients. This is consistent with findings in colorectal and pancreatic cancers, where EGFR overexpression has been linked to poor prognosis and an increased risk of metastasis [11,27]. However, while EGFR-positive patients exhibited a trend towards inferior DFS, this trend did not reach statistical significance. This lack of statistical significance may be attributed to the limited sample size, with a majority (87.4%) of patients classified as EGFR-positive. Additionally, a considerable proportion (12 of 52, 23.1%) of recurrent cases experienced local recurrence exclusively, indicating that advanced tumor stage may have exerted the most substantial influence on disease recurrence. In patients who received systemic chemotherapy after recurrence, no prognostic differences based on EGFR expression were observed. This likely results from most EGFR-negative patients not experiencing recurrence and thus not receiving systemic treatment. Validation with a larger sample size or redefining EGFR positive expression criteria is needed for further analysis.
These findings align with previous research indicating that HER2 overexpression in biliary tract cancer correlates with reduced DFS, OS, and PFS in patients receiving palliative gemcitabine with cisplatin treatment [28]. Furthermore, HER2 overexpression is recognized as a negative prognostic factor and a negative predictive biomarker for anti-EGFR treatment in metastatic colorectal cancer, reinforcing its role in poor outcomes across different histologic types [29]. Although AAC patients with HER2 overexpression did not show a significant association with survival outcomes, possibly because of their small numbers, the potential utility of HER2-targeted therapeutic strategies in this subpopulation warrants further investigation.
Comment 4: Criteria for positive EGFR and c-Met expression are not clearly defined. Consider providing more detailed scoring methods and discussing potential variability.
Response 4: We sincerely appreciate your thorough review. As the reviewer correctly pointed out, we only classified EGFR and c-Met as either negative or positive without clearly specifying the criteria for this classification. In our study, we defined positive expression of EGFR and c-Met as having an intensity of 1+ or greater, with membranous staining in 10% or more of cells. We have now added these criteria to the methods section and included representative images of each protein expression in the supplementary material. However, even if we were to further elaborate on the criteria for positive expression in IHC, the determination of protein expression positivity ultimately becomes quite complex due to factors such as the cut-off for protein expression and its correlation with the effectiveness of targeted therapy. Just as HER2 IHC positivity in breast and stomach cancers has been defined through IHC methods based on its association with the efficacy of anti-HER2 targeted treatments, it appears that further diverse discussions, including those involving other cancer types, are necessary to define positive expression of EGFR and c-Met. We have acknowledged the potential limitations of our IHC scoring method in the limitations section of our paper.
The manuscript is revised as follows:
Staining for HER2 was classified as negative (scores of 0 and 1+), equivocal (2+), or positive (3+), whereas staining for EGFR and c-Met was classified as either negative (0) or positive (any level of membranous staining greater than 10%, including scores of 1+, 2+, or 3+) (Figure S1). When there was a discrepancy in diagnosis, two pathologists discussed it to reach an agreement.
Figure S1. Representative images of EGFR, HER2, and c-Met with corresponding immunohistochemical scores ranging from 0 to 3+ (scale bar, 100 μm). The red box delineates the scores that were classified as positive for the purpose of this analysis.
This study has several limitations. First, protein expression was verified solely through immunohistochemical analysis, without additional validation via in situ hybridization (ISH) or next-generation sequencing (NGS) for gene amplification. Consequently, there may be potential discrepancies between protein overexpression and gene amplification. Moreover, as expression intensity for EGFR and c-Met was not scored, further discussion is needed to refine the definition of positive expression based on intensity.
Comment 5: Assessment of protein expression using IHC alone is controversial due to potential variability in interpretation and lack of quantitative precision. Consider discussing the limitations of using IHC alone and recommend complementary techniques (e.g., in situ hybridization, next-generation sequencing) for confirmation.
Response 5: We are deeply grateful for your comprehensive review. We acknowledge the limitation you've highlighted regarding our reliance on IHC alone for protein expression evaluation, without genetic-level validation of gene amplification. Our initial plan was to conduct a thorough examination of molecular alterations using next-generation sequencing, aiming to confirm EGFR, ERBB2, and MET gene amplification, as well as identify co-altering mutations such as KRAS. Regrettably, we encountered obstacles with older specimen quality and faced budget limitations, which prevented us from pursuing these additional analyses.
Despite our use of IHC as the sole evaluation method, it's worth noting that existing research suggests a potential correlation between protein expression detected by IHC and gene amplification. We intend to incorporate a discussion of this relationship in our manuscript to provide a more nuanced perspective. We sincerely appreciate your detailed feedback, which has been invaluable in improving our work.
The manuscript is revised as follows:
This study has several limitations. First, protein expression was verified solely through immunohistochemical analysis, without additional validation via in situ hybridization (ISH) or next-generation sequencing (NGS) for gene amplification. Consequently, there may be potential discrepancies between protein overexpression and gene amplification. Moreover, as expression intensity for EGFR and c-Met was not scored, further discussion is needed to refine the definition of positive expression based on intensity. Nonetheless, EGFR protein expression via IHC has been strongly associated with gene amplification in lung adenocarcinoma and breast cancer when detected by fluorescence and chromogenic ISH, respectively [32,33]. Similarly, in gastric cancer, MET mRNA expression through RNA ISH aligns well with c-Met protein expression by IHC [34]. However, in lung cancer, c-Met IHC shows poor correlation with fluorescence ISH and NGS results [35]. This indicates that the reliability of c-Met IHC may vary by cancer type, highlighting the need for complementary molecular validation in future research. Second, a comprehensive analysis of genetic alterations that could confer resistance to anti-EGFR and anti-HER2 targeted treatments, such as KRAS mutations, was not conducted. Further research is necessary to closely examine the relationship between these molecular alterations and prognosis. Lastly, the analysis relied solely on surgical tissue samples, without additional biopsies for recurrent patients. This limitation could introduce tumor heterogeneity as a confounding factor in the survival outcomes of these patients.
Comment 6: The English language used throughout the manuscript needs some improvements in terms of style and grammar ( for example-Table 1 Missig data )
Response 6: Thank you for your thorough comments. We have revised the manuscript to address the noted issues and improve overall clarity and grammar. If there are any additional areas that require further revision, please let us know.
Reviewer 2 Report
Comments and Suggestions for Authors
In the manuscript „Prognostic Significance of EGFR, HER2, and c-Met Overexpression in Surgically Treated Patients with Adenocarcinoma of the Ampulla of Vater“ Park et al. determined the expression status of EGFR, HER2 and c-Met in patient samples with ampulla of Vater carcinoma (AVC) and correlated the expression with the clinicopathological features and prognosis. The EGFR overexpression was observed in most patients tested and positively correlated with the worst clinical outcome. These results indicate the potential of using targeted therapy for the treatment of EGFR-positive AVC.
The topic is very interesting and the study is nicely presented. I acknowledge the authors for pointing the limitations of the study, nevertheless several comments should be addressed before considering manuscript for publication.
The authors used same sample cohort as in their already published study (PMID: 37783755) and therefore the relevant study should be mentioned in both Materials and Methods and Results Sections.
The authors based their findings on IHC results, thus the representative images of positive and negative EGFR, HER2 and c-Met expression for each AVC subtype should be shown.
Finally, it is known that KRAS mutations are one of the most frequent molecular alterations in AVC which can prevail in pancreato-biliary subtype, so I would recommend the authors, if possible, to check the KRAS mutation status in their AVC cohort and correlate the obtained findings with the expression status of EGFR, HER2 and c-Met.
Author Response
Reviewer Comments:
Reviewer 2
In the manuscript “Prognostic Significance of EGFR, HER2, and c-Met Overexpression in Surgically Treated Patients with Adenocarcinoma of the Ampulla of Vater” Park et al. determined the expression status of EGFR, HER2 and c-Met in patient samples with ampulla of Vater carcinoma (AVC) and correlated the expression with the clinicopathological features and prognosis. The EGFR overexpression was observed in most patients tested and positively correlated with the worst clinical outcome. These results indicate the potential of using targeted therapy for the treatment of EGFR-positive AVC.
The topic is very interesting, and the study is nicely presented. I acknowledge the authors for pointing the limitations of the study, nevertheless several comments should be addressed before considering manuscript for publication.
Comment 1: The authors used same sample cohort as in their already published study (PMID: 37783755) and therefore the relevant study should be mentioned in both Materials and Methods and Results Sections.
Response 1: Thank you for your invaluable feedback. As the reviewer rightly pointed out, the cohort in this study indeed matches the one from our previously published research. This study presents additional analyses of druggable protein expression within the same cohort. We completely agree that this point should be explicitly mentioned in both the methods and results sections. Accordingly, we have removed any unnecessarily repetitive expressions and have clearly articulated the additional analytical methods employed in this study using the same cohort. We sincerely appreciate your astute observation and guidance, which have helped us improve the clarity and accuracy of our manuscript.
The manuscript is revised as follows:
2.1. Patients and Data Acquisition
In this study, we analyzed a cohort of patients with surgically resected adenocarcinoma of the AAC that has been previously reported in a published study [23]. The current analysis focuses on evaluating the expression of proteins such as EGFR, HER2, and c-Met in the same patient cohort. The inclusion and exclusion criteria for this study were identical to those in the previously published study. Patients eligible for inclusion were required to meet the following criteria: (1) histologically confirmed adenocarcinoma of the ampulla of Vater; (2) pathological stage IB-III as per the 8th edition of the American Joint Committee on Cancer Staging; and (3) verifiable recurrence and survival data at the time of collection. Exclusion criteria were: (1) pathological tumor, node, metastasis (TNM) stage IA disease; (2) absence of examined regional lymph nodes; (3) presence of macroscopically residual tumors (R2 resection); (4) receipt of preoperative chemotherapy or radiotherapy; or (5) diagnosis of a secondary malignancy post-surgery.
3.1. Patients Characteristics
From August 1, 2007, to December 31, 2021, a total of 87 patients were included in this study. The demographic and clinicopathological characteristics of these patients, previously reported in an earlier study [23], are summarized in Table 1. These patients with AAC were categorized into PB type (n=54) and intestinal type (n=33). Both sub-types had similar median ages, gender distributions, mean tumor sizes, and histologic grading. The intestinal subtype had a higher proportion of T1-2 tumors compared to the PB subtype (66.7% vs. 46.3%; p = 0.064), although node category and TNM stage distributions were similar (p = 0.380 and p = 0.480, respectively). EGFR expression was significantly higher in the PB type (94.4% vs. 75.8%; p = 0.018), whereas HER2 and c-Met expressions did not show significant differences (p = 0.571 and p = 0.467, respectively).
Comment 2: The authors based their findings on IHC results, thus the representative images of positive and negative EGFR, HER2 and c-Met expression for each AVC subtype should be shown.
Response 2: We are truly grateful for your thorough review. As you have astutely pointed out, this study divides the subjects into two groups based on protein expression and examines the characteristics and prognostic correlations accordingly. Therefore, we completely agree that an objective assessment of protein expression as positive or negative is of paramount importance. Considering of this, we would like to submit additional representative images that illustrate the positive and negative protein expression as determined by IHC in our study. We believe these images will provide clarity and support for our methodology. If you think it would be more appropriate to include the representative images in the main article rather than in the supplementary material, please let us know, and we will make the necessary adjustments.
The manuscript is revised as follows:
Staining for HER2 was classified as negative (scores of 0 and 1+), equivocal (2+), or positive (3+), whereas staining for EGFR and c-Met was classified as either negative (0) or positive (any level of membranous staining greater than 10%, including scores of 1+, 2+, or 3+) (Figure S1). When there was a discrepancy in diagnosis, two pathologists discussed it to reach an agreement.
Figure S1. Representative images of EGFR, HER2, and c-Met with corresponding immunohistochemical scores ranging from 0 to 3+ (scale bar, 100 μm). The red box delineates the scores that were classified as positive for the purpose of this analysis.
Comment 3: Finally, it is known that KRAS mutations are one of the most frequent molecular alterations in AVC which can prevail in pancreato-biliary subtype, so I would recommend the authors, if possible, to check the KRAS mutation status in their AVC cohort and correlate the obtained findings with the expression status of EGFR, HER2 and c-Met.
Response 3: We are deeply grateful for your insightful comments. As reported in the literature, KRAS mutations occur in approximately 40-65% of ampulla of Vater adenocarcinoma patients. As you correctly pointed out, this mutation is even more prevalent in the PB subtype, occurring in 65-70% of cases, which is more frequent than in the intestinal subtype. EGFR, HER2, and c-Met are all cell membrane proteins. If a KRAS mutation, which is downstream in the signaling pathway, is present regardless of the overexpression of these proteins, the effectiveness of anti-EGFR, anti-HER2, or anti-Met targeted treatments would likely be diminished. Therefore, we fully agree that additional analysis of this aspect is crucial. In fact, considering the possibility of poor prognosis when mutations in the MAPK pathway (including RAS and RAF), the PI3K/Akt pathway, and TP53 are present, we initially aimed to comprehensively examine molecular alterations through next-generation sequencing. However, due to challenges with older specimens, sample quality issues, and budget constraints, we were unable to proceed with these additional tests. We fully recognize the importance of understanding how additional molecular alterations like KRAS mutations correlate with prognosis in patients with EGFR, HER2, and c-Met overexpression. We will explicitly state in the manuscript that further research in this area is necessary. If possible, in the future, we plan to conduct additional analyses, potentially using PCR-based tests to at least confirm RAS and RAF mutation status. We sincerely appreciate your valuable suggestion, which has highlighted an important avenue for future research in our study.
The manuscript is revised as follows:
This study has several limitations. First, protein expression was verified solely through immunohistochemical analysis, without additional validation via in situ hybridization (ISH) or next-generation sequencing (NGS) for gene amplification. Consequently, there may be potential discrepancies between protein overexpression and gene amplification. Moreover, as expression intensity for EGFR and c-Met was not scored, further discussion is needed to refine the definition of positive expression based on intensity. Nonetheless, EGFR protein expression via IHC has been strongly associated with gene amplification in lung adenocarcinoma and breast cancer when detected by fluorescence and chromogenic ISH, respectively [32,33]. Similarly, in gastric cancer, MET mRNA expression through RNA ISH aligns well with c-Met protein expression by IHC [34]. However, in lung cancer, c-Met IHC shows poor correlation with fluorescence ISH and NGS results [35]. This indicates that the reliability of c-Met IHC may vary by cancer type, highlighting the need for complementary molecular validation in future research. Second, a comprehensive analysis of genetic alterations that could confer resistance to anti-EGFR and anti-HER2 targeted treatments, such as KRAS mutations, was not conducted. Further research is necessary to closely examine the relationship between these molecular alterations and prognosis. Lastly, the analysis relied solely on surgical tissue samples, without additional biopsies for recurrent patients. This limitation could introduce tumor heterogeneity as a confounding factor in the survival outcomes of these patients.
Round 2
Reviewer 1 Report
Comments and Suggestions for Authors
I am satisfied that the authors have addressed all of my previous concerns about the article. It is now much improved and I feel that it is now suitable for publication.
Comments on the Quality of English LanguageI am satisfied that the authors have addressed all of my previous concerns about the article. It is now much improved and I feel that it is now suitable for publication.
Reviewer 2 Report
Comments and Suggestions for Authors
I thank the authors for satisfactorily addressing all my remarks and therefore, to my opinion, the manuscript in the revised form meets the standards of Cancers.